# Lipid Self-Assemblies and Nanostructured Emulsions for Cosmetic Formulations

**Chandrashekhar V. Kulkarni**

Centre for Materials Science, School of Physical Sciences and Computing, University of Central Lancashire, Preston PR1 2HE, UK; cvkulkarni@uclan.ac.uk; Tel.: +44-1772-89-4339; Fax: +44-1772-89-4981

**Abstract:** A majority of cosmetic products that we encounter on daily basis contain lipid constituents in solubilized or insolubilized forms. Due to their amphiphilic nature, the lipid molecules spontaneously self-assemble into a remarkable range of nanostructures when mixed with water. This review illustrates the formation and finely tunable properties of self-assembled lipid nanostructures and their hierarchically organized derivatives, as well as their relevance to the development of cosmetic formulations. These lipid systems can be modulated into various physical forms suitable for topical administration including fluids, gels, creams, pastes and dehydrated films. Moreover, they are capable of encapsulating hydrophilic, hydrophobic as well as amphiphilic active ingredients owing to their special morphological characters. Nano-hybrid materials with more elegant properties can be designed by combining nanostructured lipid systems with other nanomaterials including a hydrogelator, silica nanoparticles, clays and carbon nanomaterials. The smart materials reviewed here may well be the future of innovative cosmetic applications.

**Keywords:** lipid self-assembly; lipid nanostructures; cosmetics; formulations; oil-in-water emulsions; water-in-oil emulsions; hierarchically ordered materials; nano-hybrids

## 1. Introduction

The cosmetic product, according to the EU regulation, is defined as "any substance or mixture intended to be placed in contact with the external parts of the human body (epidermis, hair system, nails, lips and external genital organs) or with the teeth and the mucous membranes of the oral cavity with a view exclusively or mainly to cleaning them, perfuming them, changing their appearance, protecting them, keeping them in good condition or correcting body odours" [1]. There are about 25,000 ingredients that constitute cosmetic products, most of which can be grouped into oils, waxes, and fats—broadly defined as "lipids" [2]. Natural sources (e.g., plants and animal) still appear to be the preferred means of obtaining these ingredients over synthetic derivatives [2].

Lipids form an integral part of cosmetic products [3–6]. Their cosmetic applications are fundamentally determined by their physico-chemical properties. For instance, due to hydrophobic nature, they can act as a barrier against water-loss from the skin. Their viscosity and semi-solid consistency defines the spreadability, which is an important factor to fill small indentations on the skin surface imparting smooth and shiny appearance. Not only the melting point, but also the phase behavior of a lipid is crucial for the stability and performance of the cosmetic product. Lipids form a remarkable range of self-assembled nanostructures exhibiting very different architectures and properties to each other. Therefore, if the lipid phase behavior at a range of temperatures and water contents is known, it is possible to control the properties and performance of the cosmetic products during their processing, storage, transport and final applications. This review describes the types of lipid nanostructures, their derivatives with improved performance and possible ways to fine-tune their properties that can influence the development of cosmetic formulations.

## 2. Lipid Self-Assembly: Simple and Elegant Nanostructures

Among over 40,000 unique lipid structures known to date, fatty acyls and glycerides comprise more than half [7] (Figure 1). These lipids are also common for an extensive range of cosmetic products [2]. Lipids are generally *amphiphilic* in nature—*amphi* means two and *philic* means liking or loving. The lipid molecule has two parts: the hydrophilic head group, which likes water and long alkyl chain/s, which do not like water. This special chemistry governs the self-assembling process of lipids when mixed with an aqueous medium [8–11].

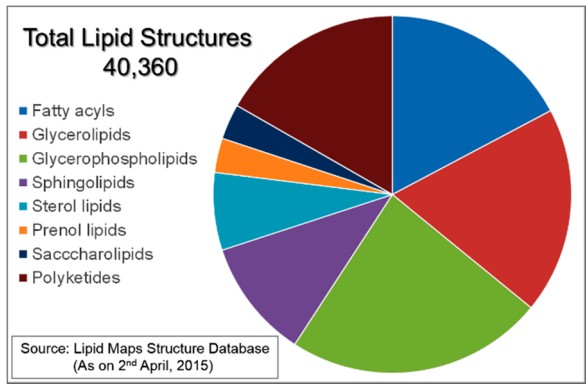

**Figure 1.** Main categories of lipid structures deposited on Lipid Maps Structure Database [7].

An average shape of the amphiphilic molecule determines the type of self-assembly (Figure 2) [11,12]. Molecules with large hydrophilic groups, e.g., detergents, tend to form normal (type 1) self-assemblies while rather small head group molecules, e.g., lipids, have a tendency to form inverse (type 2) phases. Biological membranes constitute a large assortment of different shaped molecules having a tendency to form dynamic self-assemblies including locally planar structures (type 0). Spherical micelles and cylindrical micelles are simple self-assemblies formed in both normal and inverse types, whereas lamellar phase is formed when the molecule acquires straight (average) shape (Figure 2). Vesicles (liposomes) are also formed in this situation, as, at local level, they exhibit planar lamellar phase.

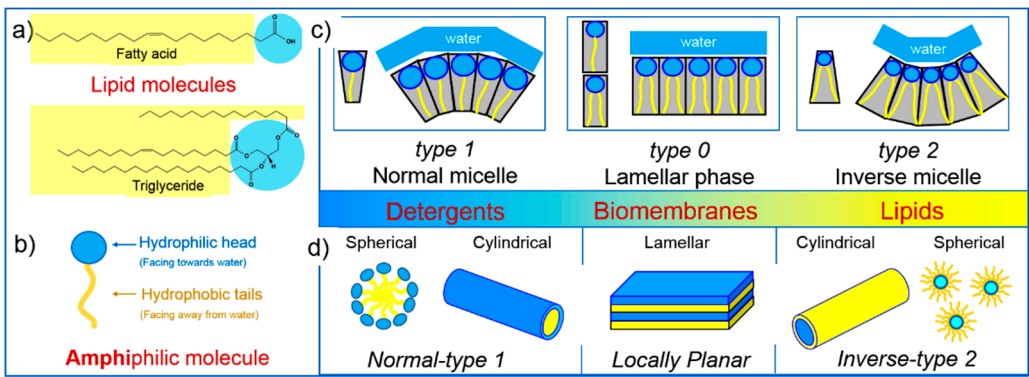

**Figure 2.** (**a**) Chemical structures of typical lipid molecules; (**b**) Schematic diagram of an amphiphilic molecule where blue and yellow colors roughly code for hydrophilic and hydrophobic parts of the molecule; (**c**) Average molecular shapes and process of self-assembling of amphiphilic molecules when they are mixed with water; (**d**) Schematics of simple self-assemblies formed by amphiphilic molecules.

Vesicles, also called liposomes, are formed when a single or multiple lipid bilayers organize into a spherical shape to enclose the water (Figure 3) [13]. However, the shape is not always spherical;

vesicles take a range of shapes and are hence given corresponding names, such as prolate or starfish vesicles [14]. If the vesicle is formed of a single bilayer, it is called a uni-lamellar vesicle (ULV). Similarly, if it constitutes multiple bilayers, it is called a multi-lamellar vesicle (MLV) (Figure 3). Sometimes, a large vesicle engulfs similar or different sized smaller vesicles, it is called oligo-vesicular vesicle (OVV) [14]. Based on their size uni-lamellar vesicles are broadly categorized into small (SUV, 10–100 nm), large (LUV, 100–1000 nm) and giant (GUV, >1 μm) uni-lamellar vesicles [15] (Figure 3). These vesicles are highly popular among biotechnological researchers due to their prominent applications, including model membranes [16] and drug delivery carriers [17].

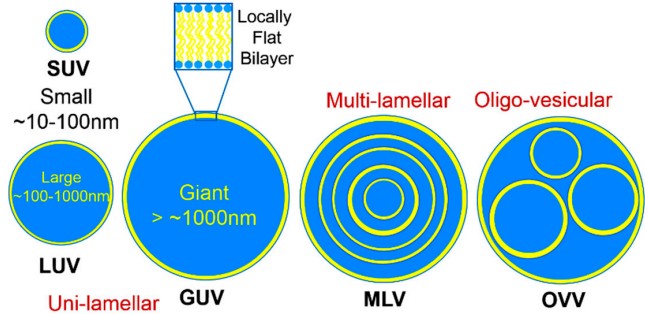

**Figure 3.** Types and schematic diagrams of vesicles (liposomes).

Much more elegant and complex morphologies are formed by lipids under various environmental factors in zero-, one-, two- and three-dimensional space (Figure 4). The spatial arrangement of these lipid phases mediates crystalline solids and Newtonian liquids, hence they are often termed as liquid crystalline (LC) phases. However, they are getting more recognition as "nanostructures" due to several representative properties: their unit cell sizes scale in the range of 0.25–35 nm [15,18] and the lipid bilayer thickness for cubic phases lies in 2.0–6.2 nm length scale [18]; whereas aqueous channel diameters for bicontinuous cubic phases can be anywhere between 1.0 nm and 13.0 nm [18].

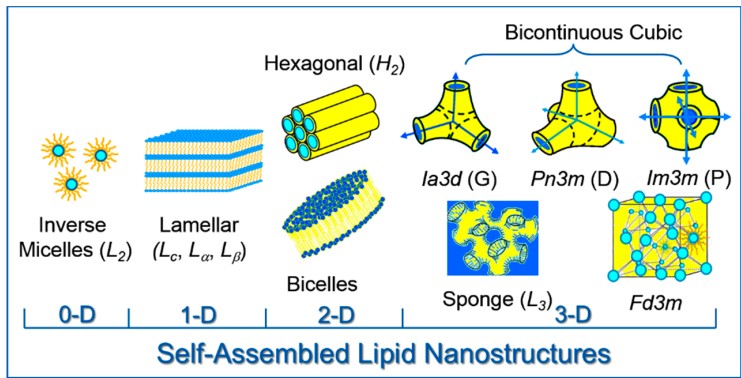

**Figure 4.** Types and schematic diagrams of thermodynamically stable self-assembled lipid nanostructures.

In dry state or low hydration and/or temperatures, lipids tend to form crystalline lamellar phase ($L_c$) due to restricted molecular motion. With further increase in temperature and/or hydration they may modify into various polymorphs, for instance, lamellar gel ($L_\beta$), fluid lamellar ($L_\alpha$) or simply inverse micelles ($L_2$) [8,19]. The lamellar phase is usually formed by one-dimensional (1-D) stacking of lipid bilayers separating adjacent hydrophilic (water) layers; each such bilayer is formed by two back-to-back monolayers shielding the hydrocarbon chains away from water (Figure 4). Lipids in stratum corneum of human skin form highly ordered layers adopting lamellar nanostructures [20]. Small angle X-ray diffraction studies revealed the lamellar repeat spacing in the range of 60–65 Å,

while the lipid head group packing displayed crystalline and fluid lamellar polymorphs with wide angle reflections at 4.0–4.2 Å (many sharp reflections) and 4.6 Å (broad reflection), respectively [20]. This structured lipid layer helps skin to withstand desiccation and thermal regulation. The triglyceride triacylglycerol (TAG) lipids exhibit additional range of polymorphism (further types of crystalline and gel polymorphs) generally seen in the structure of foods [21,22]. TAGs are very common lipids employed in cosmetic formulations [3,6], displaying a broad lamellar phase behaviour discussed above.

The inverse hexagonal phase ($H_2$) is a common non-lamellar phase formed by many lipids [10]. It consists of aqueous conduits enclosed within two-dimensional (2-D) cylindrical lipid architectures (Figure 4). The $H_2$ phase formed during the transient re-assembly of lipids in stratum corneum is presumed to form an effective barrier against water transport similar to lamellar nanostructures [20]. The bicelle is a 2-D nanostructure formed from a small bilayer taking the shape of a disk (Figure 4). It is usually formed by a mixture of more than one lipid (or similar) components [23]. The structure of the bicelle mediates the morphology of vesicles and conventional mixed micelles, thereby offering an attractive model membrane properties [23].

Three-dimensional (3-D) and elegant nanostructures include following lipid phases: bicontinuous cubic phases, sponge phase and micellar cubic phases (Figure 4). Bicontinuous cubic phases are geometrically well defined, based on mathematically minimal surfaces of primitive (P), diamond (D) and gyroid (G) type [24]. The continuous bilayer is draped around these surfaces separating two aqueous networks, forming corresponding cubic phases with crystallographic groups of Im3m, Pn3m and Ia3d [24,25] (Figure 4). These phases are called "bi"-continuous owing to their continuity in aqueous as well as lipid bilayer networks. The sponge phase ($L_3$) is also bicontinuous in nature but the bilayer is not well-ordered as in cubic phases [26–28] (Figure 4). Bicelles [29], bicontinuous cubic [30] and sponge [31] phases are very useful for biotechnological applications, for instance, for reconstitution and crystallization of membrane proteins. But more interestingly, bicontinous cubic and hexagonal phase structures have been recognized as "biomimetic" as they were seen in sub-cellular membranes of endoplasmic reticulum, Golgi and mitochondria [32,33]. Micellar phases are formed of discrete micelles ordered on a regular lattice [34,35]. The Fd3m phase is an example of micellar cubic phase consisting of two different sized micelles ordered on a cubic lattice [36] (Figure 4). Similar to bicelles, this phase is also adopted by a mixture of components rather than pure lipid-water systems [36]. Above lipid nanostructures are categorized as thermodynamically stable phases under equilibrium conditions. Most common lipid phases are discussed above, however there are some other phases—either intermediate or metastable and not commonly found (for details, readers are advised to refer corresponding publications [27,37–40]).

Topical applications of various bulk self-assembled lipid nanostructures have been successfully demonstrated by several research groups [41–45]. On conventional lipid-water phase diagrams, the bicontinuous cubic phases are generally observed at higher water fractions as compared to lamellar phases [46–48]. This also means that cubic phases show stronger water holding capacity than lamellar nanostructures. It was experimentally demonstrated that the cubic phase containing gel increases water binding to the skin [49]. On the contrary, the cubic phase was shown to be highly vapor permeable leading to more trans-epidermal water loss (TEWL) compared to the lamellar phase applied on human skin [49]. This may be attributed to the open network of water channels in bicontinuous cubic phase. Moreover, the cubic phases are highly viscous and sticky, making them difficult to handle and difficult to apply on skin, whereas the rather anhydrous lamellar phase is fluid and thus easy to apply. Nonetheless, the handling difficulties of viscous phases can be reduced by simply dispersing them into fluid emulsion forms as described in the following section.

## 3. Lipid Nanostructured Emulsions: Next Level of Structural Hierarchy

Self-assembled lipid nanostructures including lamellar, hexagonal and cubic phases are potentially useful for encapsulation of functional molecules [50–57]. Their sustained release applications have been reported recently [51,58–61]. However, these phases, especially cubic phases exhibit very high

viscosity [62–65], causing difficulties in their precise harvesting and thus handling using conventional techniques. One of the best ways to enhance the applicability of these lipid nanostructures is to disperse them into particulate or less viscous systems [66–68]. The resulting systems display hierarchically ordered morphologies ranging from nanometer to millimeter length scales; for example see [68].

Dispersed systems are usually some form of emulsion; for instance, oil-in-water (O/W) or water-in-oil (W/O) emulsions of lipid components in an aqueous medium (Figure 5). The O/W emulsions of lipid nanostructures are usually prepared by dispersing lipid (or lipid additive mixture) in an aqueous solution of a stabilizer molecule. For this, the above mixture is subjected to a high energy input, such as ultra-sonication, by which highly viscous liquid crystalline phase is fragmented into discrete particles of micron or sub-micron size [66]. Interface of these particles is then kinetically stabilized by stabilizer molecules keeping them dispersed and thus preventing their further aggregation. The resulting dispersion exhibits water-like fluidity while maintaining self-assembled nanostructures at the cores of dispersed particles. These particles are internally self-assembled, hence they are also called "isasomes" ("somes" here refers to particles) [69]. More specifically, the particles with the cubic self-assembly are called "cubosomes" [66], while "hexosomes" are the particles having an interior of the hexagonal type [70]. It is possible to modulate the internal self-assembly into other nanostructures including lamellar phase, sponge phase, micellar cubic phase and inverse micelles [15]. This is an important distinguishing feature of these nanostructured emulsions from conventional O/W emulsions where the oil phase is simply made up of oily droplets—usually fluid and disordered. However, the oil phase in nanostructured emulsions is usually a semi-solid liquid crystalline phase, which is highly ordered in nanometer length scales.

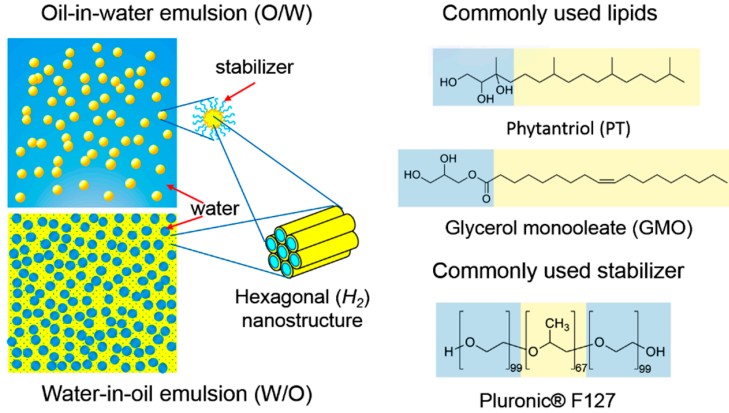

**Figure 5.** Kinetically stabilized oil-in-water (O/W) and water-in-oil (W/O) nanostructured emulsions comprising a self-assembled lipid nanostructure are shown in the left panel. The hexagonal nanostructure is shown as an example but it should be noted that both of these emulsions can be prepared with other types of nanostructures, including bicontinuous cubic Pn3m or Im3m, micellar cubic Fd3m, hexagonal ($H_2$) or inverse micelles ($L_2$). Chemical structures of the most common lipids and a stabilizer used to prepare nanostructured emulsions are shown on the right.

The primary components of O/W nanostructured emulsions are: lipid, stabilizer and water (Figure 5). The lipids, namely monoolein (MO) and monolinolein (ML), have been utilized in their pure forms or as commercial grades, glycerol monooleate (GMO) and dimodan (DU), respectively. Phytantriol (PT), which is a well-known cosmetic ingredient [71], is becoming popular in preparing nanostructured particle dispersions [60,72–74]. Some other lipids exploited for the preparation of cubosomes include, monoelaidin, myverol, Soy phosphatidylcholine/glycerol dioleate and 1,2-dioleoyl-sn-glycero-3-phosphocholine (DOPC)/1,2-dioleoyl-sn-glycero-3-phosphoethanolamine (DOPE) [74–77]. Recently we have prepared O/W emulsion using a food material, butter oil, which is a cheap and abundant source of lipid molecules [78].

A wide range of stabilizers [76] have been used for O/W emulsions including surfactants [66], silica nanoparticles [79], clay platelets [80] and carbon nanotubes [81,82]. The pluronic F127, a triblock co-polymer has been very popular stabilizer for nanostructured emulsions among others [74,83] (Figure 5).

O/W nanostructured emulsions generally require simple preparation set-ups that can be easily scaled up for commercial production [84–86]. As the nanostructured particles are based on a lipid self-assembly, the availability of lipophilic volume and an interfacial area per particle (up to 400 m$^2$/g) is far greater than the conventional liposomes [50]. This is, in fact, quite advantageous for poorly water soluble functional molecules. O/W emulsions show great potential for encapsulating hydrophilic, hydrophobic or amphiphilic molecules, which are presumably hosted in an aqueous continuous medium, lipophilic region with self-assembled lipid nanostructures and native-like bilayer membrane, respectively [50,74,87]. It has been demonstrated that these nanostructured particles can be administrated via oral, intravenous, dermal, intradermal, mucosal, ocular, intranasal, percutaneous, intraperitoneal and intra-tympanic routes [74], which is certainly promising for developing cosmetic applications of the above systems. Moreover, the biocompatibility studies of these particles have been performed via interacting with cell membranes, model membranes [88] and blood components [89].

O/W emulsions, discussed above, are primarily the lipid self-assemblies dispersed in water taking a colloidal (particulate) form. Conversely, W/O nanostructured emulsions (Figure 5) have these phases reversed, meaning the dispersed phase is water and the continuous phase can be one of the lipid self-assemblies [68,90–93]. The type of self-assembly can be bicontinuous cubic Pn3m or Im3m, micellar cubic Fd3m, hexagonal ($H_2$) or inverse micelles ($L_2$) (Figure 5). W/O emulsions are usually very concentrated and their stability is governed by the concentration of the components [94,95]. For instance, the emulsion can be stabilized by an optimum concentration of emulsion stabilizer or by obtaining a very high concentration of an internal phase or both of the above. Emulsions containing >74% dispersed phase are called high internal phase ratio emulsions (HIPRE) or high internal phase emulsions (HIPE), or alternatively bi-liquid foams or gel emulsions [90,96–99].

Interestingly, one of the recent reports reveal that the W/O nanostructured emulsions can be prepared without using any external stabilizer [68,76,100]. Moreover, the preparation methodology discussed therein offers a range of possibilities to fine-tune them with respect to their nanostructure and properties. It was anticipated that the emulsion stability is brought by an apparent viscosity of the self-assembled nanostructure (except the emulsion with the $L_2$ phase, where the stability is supposed to be imparted by the packing of droplets, similar to HIPRE). These emulsions were prepared using a custom-made device based on high temperature shearing of pre-mixed lipophilic and hydrophilic phases [68]. However, we noticed that these emulsions can be crudely prepared by heating the mixture of lipophilic and hydrophilic phases above melting transition, followed by a rigorous mixing (e.g., using vortex) while they are cooled at room temperature. Exciting advantages of these W/O nanostructures for cosmetic formulations are as follows [68,101]: (1) they can be prepared without any stabilizer thus facilitating the development of surfactant-free cosmetic products; (2) they can imbibe 50–90 wt % water, ideally suited for an efficient moisturizing agent; (3) they can be prepared via simple lab-based techniques as well as by custom-made scalable technology; (4) they can be made to adopt a range of physical forms designated for cosmetics, for instance, fluid, creamy, paste-like, gel-like or dynamically arrested form [68]; (5) they exhibit hierarchically ordered architectures which offer a great degree of tunability at various levels; and (6) they have an ability to encapsulate functional molecules of hydrophilic, hydrophobic and amphiphilic types.

In addition to the above discussed O/W and W/O nanostructured emulsions, there are a range of other emulsions [15] including double-emulsions, nano-emulsions, micro-emulsions, poly high internal phase ratio emulsions (polyHIPRE) [102], polymerized lipid self-assemblies [103–106], bicontinuous inter-facially jammed emulsion gels (Bi-jels) [107], liquid crystalline elastomers [108], solid lipid nanoparticles (SLN) [109], and layer-by-layer assemblies involving lipid nanostructures [110]; however, these are not discussed any further.

## 4. Modulating Structure and Properties of Lipid Systems

Structure, properties and performance of lipid systems are closely linked, therefore it is possible to fine-tune the structure over a broad range and achieve desired properties and a suitable performance. For instance, the internal self-assembly of nanostructured emulsions can be altered from cubic Pn3m to cubic Im3m by adding diglycerol monooleate or pluronic F127 to a DU lipid system [111,112]. The size of lipid nanoparticles can be modulated by changing the concentration of the dispersed phase and/or the shear rate of the shearing device [94]. Nanoscale parameters can be moderated further by using a different emulsifying lipid; for instance, GMO and PT can be used for many applications where the DU is demonstrably used [60,73,113]. Addition of additives in varying proportions can induce structural changes; the additives can be an oil (tetradecane (TC), *R*(+)limonene (LM), etc.), glycerol, polysaccharides or other surfactants [50,112,114–118].

The temperature [119,120], the pressure [119], an effective charge [72,80,121] or the pH [122] can also cause structural transitions in the self-assembled nanostructures of lipid particles. The type of internal nanostructure is important because it can determine the possibility and efficiency of the encapsulation of functional molecules within the lipid system. Moreover, the release rates are strongly influenced by the type and lattice parameters of the nanostructure of lipid systems [53,60,74].

The rheological properties, especially the viscosity and the yield stress, are critical parameters while developing cosmetic products [123]. These properties differ largely among the self-assembled lipid nanostructures and their hierarchically organized derivatives as discussed below (Figure 6). Liposome solutions are usually very diluted and thus display water-like consistency. On the other hand, the viscosity (or storage modulus) of self-assembled bulk phases broadly increases from $L_\alpha$ phase to $H_2$ phase to bicontinuous cubic phases [65,124], providing them viscous gel-like consistency (Figure 6). The dispersed system of nanostructured lipid particles (O/W emulsions) exhibits rather low viscosity (milk-like consistency) (Figure 6), which is actually one of the main motivations behind their preparation. Nonetheless, these O/W emulsions can be made viscous, if required, by various means, for instance, by increasing the concentration of the dispersed phase (usually a lipid phase) [94] or by preparing a hydrogel in the continuous phase of these emulsions [125,126] (Figure 6). The latter can be employed to partially arrest the dynamics of the nanostructured lipid particles [125]. This mixed system can be subsequently dehydrated to obtain dry films where the lipid particles are fully immobilized [127] (Figure 6). These thermos-reversible hydrogel systems can be rehydrated and re-dispersed on demand, which is highly useful for their storage, transport and potential biotechnological applications [128].

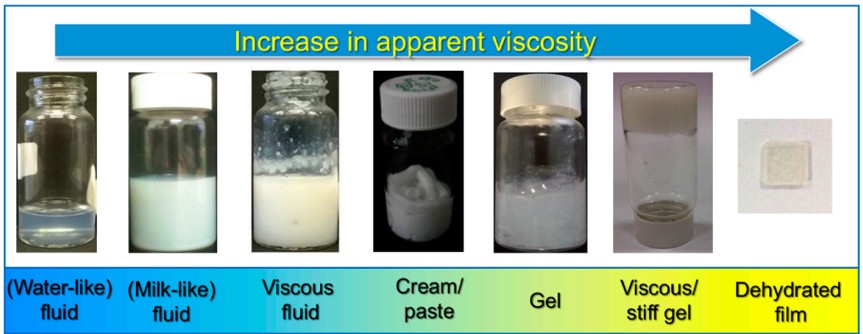

**Figure 6.** Apparent viscosity and virtual appearance of lipid-based systems. From left to right: liposome solution, O/W nanostructured emulsion, dense O/W water nanostructured emulsion, W/O nanostructured emulsion, bicontinuous cubic phase of a lipid, O/W water emulsion with a hydrogelator (does not fall due to gravity) and a film made by dehydrating the O/W water emulsion with a hydrogelator. All of these samples were prepared in the author's laboratory at different times.

The W/O nanostructured emulsions exist in a range of consistencies including fluid, gel and cream, and also offer a great tunability in terms of their apparent viscosity [68,101]. Apparent viscosity

(and yield stress) can be controlled by varying the dispersed phase content ($\phi$), the proportion of the lipid phase (given in terms of $\delta$ value) and the concentration of hydrogelator ($\varepsilon$), as illustrated in Figure 7 [68]. Other parameters including temperature, type of nanostructure and type of lipid also influence the viscoelastic behavior of these nanostructured emulsions [68,101].

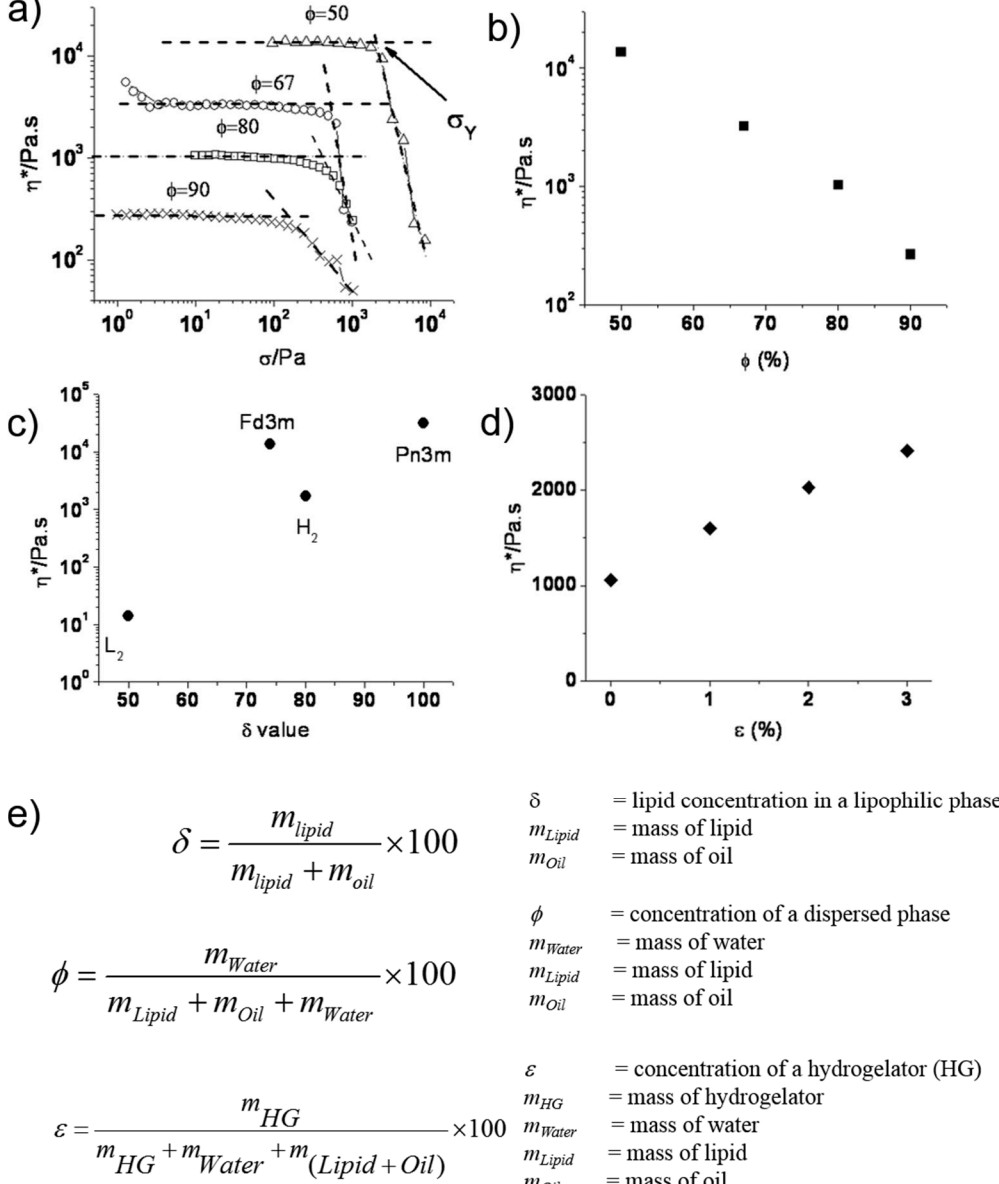

**Figure 7.** Controlling the yield stress (**a**) and the viscosity of W/O emulsions by changing the concentration of dispersed phase, $\phi$ (**a**,**b**), the $\delta$ value influencing the type of lipid nanostructure (**c**), and the concentration of hydrogelator, $\varepsilon$ (**d**); Figure modified from [68]; (**e**) Definition of terms displayed in above graphs (**a**) to (**d**). $\eta^*$ represents a complex viscosity, $\sigma$ represents shear stress and $\sigma_Y$ represents yield stress.

## 5. Nano-Hybrid Systems Based on Lipid Nanostructures

Recently, we have employed a very different approach to prepare O/W nanostructured emulsions. We developed these lipid-based systems by using a variety of carbon nanotubes (single-walled, hydroxyl-functionalized multi-walled and carboxyl-functionalized multi-walled) as stabilizers [81,82] (Figure 8). Another hybrid system was prepared from a different elegant carbon nanomaterial,

fullerene ($C_{60}$). The $C_{60}$ was incorporated in the F127-stabilized O/W nanoparticle system [129] (Figure 8). Although these hybrid nano-systems might be considered non-biocompatible, they could be potential candidates for developing novel and innovative applications in future. The apparent reason is that the toxicity of pristine carbon nanomaterials is presumed to be greatly reduced by coating them with the lipid molecules, which are already biogenic materials. Similar efforts were also made by other researchers who managed to stabilize O/W nanostructured emulsions using silica nanoparticles [72] and laponite or montmorillonite clay platelets [80,130,131]. As mentioned earlier, we were also able to construct lipid-hydrogel hybrid systems and demonstrate their potential for loading and release of functional molecules in a sustained manner [128] (Figure 8).

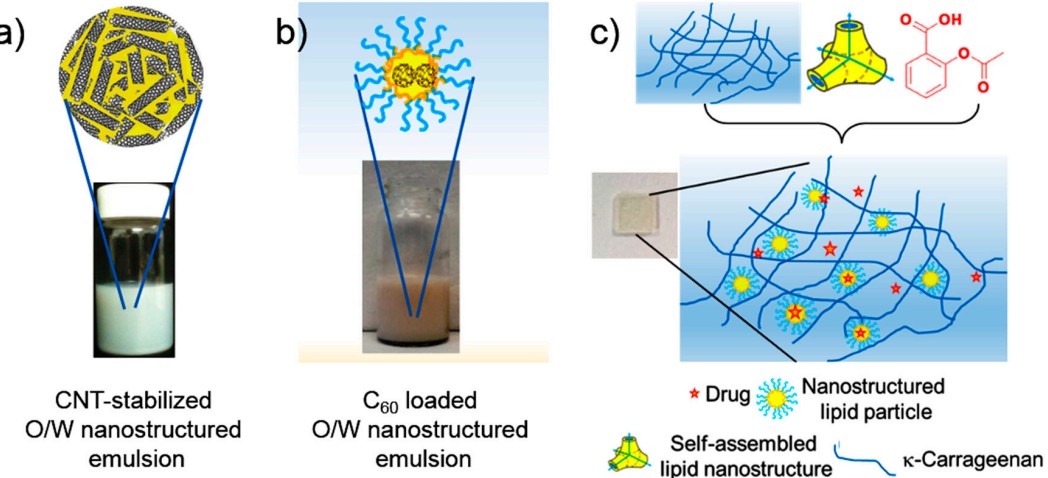

**Figure 8.** Lipid-based nano-hybrid systems prepared using (**a**) carbon nanotubes (CNTs); (**b**) fullerene ($C_{60}$) and (**c**) κ-carrageenan (KC) as a hydrogelator (and a drug, aspirin).

These novel nanocomposite systems appear to be highly promising for future cosmetic applications as they could benefit from the special characteristics of both "lipid nanostructures" and the "other component, i.e., CNT, $C_{60}$, silica nanoparticle, clay platelet or hydrogel". It is thus possible to load them with more than one active ingredient, thereby opening new avenues in terms of simultaneous delivery of multiple molecules.

## 6. Conclusions and Perspectives

This report collectively describes a range of lipid-based systems including the basic self-assembly, hierarchically organized systems, novel food-based emulsion and nano-hybrid lipid systems. The review is not comprehensive but provides a different approach to the existing lipid-based materials and sheds light on their potential development into various cosmetic products by fine-tuning their properties at various levels.

**Conflicts of Interest:** The authors declare no conflict of interest.

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
