# Peer review of "Lipid Self-Assemblies and Nanostructured Emulsions for Cosmetic Formulations"

_cosmetics, doi:10.3390/cosmetics3040037_

Round 1

Reviewer 1 Report

The review is very interesting and of great interest for all working on the cosmetic field,expecially  for the final conclusion. It should be of interest to go on and publish on Cosmetics your first studies reported as conclusion of the paper

Reviewer 2 Report

Line 51-52. References should be reported according to the journal style.

Line 270. The authors state that “the dispersed system of nanostructured lipid particles (O/W emulsions) exhibit rather low viscosity”. Generally, O/W emulsions are regarded as  dispersions of oil droplets in water. The authors should explain why they refer to O/W as a dispersion of nanostructured lipid particles instead of oil droplets.

The reference list should be checked and references should be reported consistently in accordance with the journal format.

Reviewer 3 Report

- The "nano" type for the structures discussed is not supported by numbers/sizes/comparisons. The paper advances to page 7 without justifying the "nano" assignment in the title.

- The connection to cosmetic formulations not sufficiently discussed.

- Paper contains too much background information which has been available for some time in textbooks (e.g info about o/w , w/o, lines 202-204 0n page 6, and more).

- The language used is obscure to the point of being not understandable in several places.

("This review accounts the types of lipid nanostructures..", or lines 217-220 on page 7).

Round 2

Reviewer 3 Report

N/A